# Impact of Sample Selection on In-Context Learning for Entity Extraction from Scientific Writing

**Necva Bölücü, Maciej Rybinski, Stephen Wan**
CSIRO Data61
{necva.bolucu;maciek.rybinski;stephen.wan}@csiro.au

## Abstract

Prompt-based usage of Large Language Models (LLMs) is an increasingly popular way to tackle many well-known natural language problems. This trend is due, in part, to the appeal of the *In-Context Learning* (ICL) prompt set-up, in which a few selected training examples are provided along with the inference request. ICL, a type of few-shot learning, is especially attractive for natural language processing (NLP) tasks defined for specialised domains, such as entity extraction from scientific documents, where the annotation is very costly due to expertise requirements for the annotators. In this paper, we present a comprehensive analysis of in-context sample selection methods for entity extraction from scientific documents using GPT-3.5 and compare these results against a fully supervised transformer-based baseline. Our results indicate that the effectiveness of the in-context sample selection methods is heavily domain-dependent, but the improvements are more notable for problems with a larger number of entity types. More in-depth analysis shows that ICL is more effective for low-resource set-ups of scientific information extraction.[1]

## 1 Introduction

Extracting relevant information from scientific documents plays a crucial role in improving methods for organising, indexing, and querying the vast amount of existing literature (Nasar et al., 2018; Weston et al., 2019; Hong et al., 2021). However, annotating datasets for scientific information extraction (IE) is a laborious and costly process that requires the expertise of human experts and the development of annotation guidelines.

In recent years, large language models (LLMs) have demonstrated remarkable performance on various natural language processing (NLP) tasks (Wei et al., 2022; Hegselmann et al., 2023; Ma et al.,

---

[1]The code is publicly available at https://github.com/adalin16/ICL_EE.

2023), including entity extraction from scientific documents (Dunn et al., 2022), and also for leveraging reported scientific knowledge in downstream data science applications (Sorin et al., 2023; Vert, 2023). These models, such as GPT-3 (Brown et al., 2020) and LLAMA (Touvron et al., 2023), with billions of parameters and pre-trained on vast amounts of data, have showcased impressive capabilities to tackle tasks in a zero- or few-shot learning by leveraging in-context learning (ICL) (Radford et al., 2019; Brown et al., 2020).

In ICL, models are provided with a natural language prompt consisting of three components: a format, a set of training samples (input-label pairs—demonstrations), and a test sentence. LLM outputs the predictions for a given test input without updating its parameters. The main advantage of ICL is its ability to use the pre-existing knowledge of the language model and generalise from a small number of context-specific samples. However, ICL has been shown to be sensitive to the provided samples and randomly selected samples have been shown to introduce significant instability and uncertainty to the predictions (Lu et al., 2021; Chen et al., 2022; Agrawal et al., 2022). This issue can be alleviated by optimising the selection of the in-context samples (Liu et al., 2021; Sorensen et al., 2022; Gonen et al., 2022).

ICL sample selection methods can be divided into 2 categories: (1) the methods for choosing samples from the train set (e.g., the KATE method (Liu et al., 2021)), and (2) finding the best prompts by generating samples (e.g., the Perplexity method (Gonen et al., 2022), SG-ICL (Kim et al., 2022)). These methods can significantly reduce the need for extensive human annotation and allow LLMs to adapt to various domains and tasks.

We rely on the survey of ICL (Dong et al., 2022) and delimit the methods for sample selection, from the inference stage of ICL. Our aim is to provide a comprehensive analysis of these methods for se-

lecting samples from the train set as part of ICL for *Entity Extraction* from scientific documents. Most of the methods have been applied with prompt generation (i.e., to select the best *generated* sample). Here, we use the methods only for sample selection from the training set of the dataset for entity extraction from scientific documents and compare their effectiveness for this problem. We also propose the use of the Influence method (Koh and Liang, 2017) in an oracle setting, to provide a best-case scenario to compare against. We investigate the in-context sample selection methods (see §3) and evaluate the methods adapted for entity extraction problem on 5 entity extraction datasets: ADE, MeasEval, SciERC, STEM-ECR, and WLPC, each covering a different scientific subdomain or text modality (see §4.1 for dataset overview).

Our experiments show that while fully supervised finetuned PLMs are still the gold standard when training data can be sourced, choosing the right samples for ICL can go a long way in improving the effectiveness of ICL for scientific entity extraction (see §5.1). Our experiments demonstrate an improvement potential of 7.56% on average across all experiments, when comparing the oracle method (the Influence method) to the random sample selection baseline, and 5.26% when using the best-performing method in a test setting (KATE). Moreover, our evaluations show that our main conclusions hold in a simulated low-resource setting (see §5.2). Finally, our extensive experiments allow us to synthesise some prescriptive advice for other NLP researchers and practitioners tackling scientific entity extraction (see § 5.5).

## 2  Related Work

By increasing the size of both the model and the corpus, LLMs have demonstrated the capability of ICL, which uses pre-trained language models for new tasks without relying on gradient-based training (Brown et al., 2020). In various tasks, such as inference (*ibid*), machine translation (Agrawal et al., 2022), question answering (Huang et al., 2023; Shi et al., 2023), table-to-text generation (Liu et al., 2021) and semantic parsing (An et al., 2023), the ICL use of LLMs mentioned by Brown et al. (2020) has been shown to be on par with supervised baselines in terms of effectiveness.

Other studies have found, however, that ICL does not always lead to better results than fine-tuning. Previous studies investigating ICL for IE

are very limited (Gutiérrez et al., 2022; Wan et al., 2023). Gutiérrez et al. (2022) evaluate the performance of ICL on biomedical IE tasks, Named Entity Recognition (NER) and Relation Extraction (RE). In addition, Wan et al. (2023) apply an entity-aware demonstration using the $k$NN sample selection method (Liu et al., 2021) for RE.

To the best of our knowledge, our work is one of the first attempts for IE from scientific documents that present a comprehensive analysis of in-context sample selection methods for the problem with detailed analysis.

## 3  Methods

In this section, we describe the ICL sample selection methods for entity extraction from scientific documents. First, we describe the ICL approach in Section 3.1 and then introduce the sample selection methods in Section 3.2.

### 3.1  In-context Learning

Given an LLM, ICL can be used to solve the entity extraction problem for $D = (X, Y)$, where X are the sentences ($s = w_1, \cdots, w_n$) and Y are the entities for each sentence. The prompt $P$ consists of $k$, the number of samples for the few-shot learning, samples ($T$) (selected from the train set or generated; in this work, we focus only on the former) with gold entities ($T(s_l^{train}, e_l^{train})$ is the $l^{th}$ sample) with a format ($I$) and a test sentence ($s_i$) ($P = I + T + s_i^{test}$) (see Appendix B). Prediction is done by selecting the entities with the highest probability for each sentence in the test set.

### 3.2  Sample Selection Methods

We follow the survey in-context learning (Dong et al., 2022) and choose the following methods to use for sample selection for ICL entity extraction from scientific documents.

**KATE (Knn-Augmented in-conText Example selection)**  is a $k$NN-based method to select $k$ samples which are close to test sample based on sentence embeddings and distance metrics (Euclidean or Cosine Similarity). We follow KATE to select samples from the train set of datasets for each sentence in the test set.

**Perplexity**  is a metric to evaluate the performance of language models by calculating the probability distribution of the next token given the content provided by the preceding tokens. The metric

|  |  | ADE | MeasEval | SciERC | STEM-ECR | WLPC |
|---|---|---|---|---|---|---|
| **Train set** | # Sentences | 3,076 | 542 | 1,861 | 942 | 8,581 |
|  | # Tokens | 65,244 | 18,642 | 45,412 | 20,801 | 108,047 |
|  | # Entities | 7,768 | 882 | 5,568 | 4,560 | 25,229 |
| **Dev set** | # Sentences | 769 | 155 | 275 | 118 | 2,859 |
|  | # Tokens | 16,715 | 6,069 | 6,521 | 2,697 | 36,490 |
|  | # Entities | 1,993 | 278 | 808 | 605 | 9,207 |
| **Test set** | # Sentences | 427 | 294 | 551 | 118 | 2,861 |
|  | # Tokens | 8,755 | 10,068 | 13,401 | 2,470 | 37,371 |
|  | # Entities | 1,069 | 499 | 1,681 | 559 | 9,707 |
|  | Avg $e$ | 15.30 | 9.16 | 19.28 | 18.59 | 6.82 |
|  | Avg $s$ | 131.75 | 171.56 | 151.35 | 146.76 | 75.28 |
|  | # Entity types | 2 | 1 | 6 | 4 | 18 |

Table 1: Statistical details of datasets. Avg $e$ is the average length of entities and Avg $s$ is the average length of sentences.

provides insights into the unexpectedness of a sentence in the context of a given language model. Gonen et al. (2022) use perplexity scores of prompts to select the best prompt, rather than selecting examples from the dataset, and synthetically generated prompts through paraphrasing with GPT-3 and back-translation. Unlike Gonen et al. (2022), in the experiments we focus on selecting in-context samples from the training set instead of selecting the better prompt. As the sample selection method, we calculate the perplexity of each train sentence using a language model (LM) and take the $k$ samples from the train set with the lowest perplexity, which means the sentence is more likely and consistent with the patterns it has learned from the training data of LM. Unlike the other in-context sample selection methods (Random, KATE, etc.), the selection of the $k$ samples is independent of the test sentences (i.e., the same samples from the train set are characterised by lower perplexity, independently from the test sample presented alongside).

**BM25** is a bag-of-words retrieval model that ranks relevant samples (sentences) appearing in each train set by relevance to a given test sample (Schutze et al., 2008; Robertson et al., 2009). Similar to retrieval-based methods for augmentation of the input with similar samples from the train set (Xu et al., 2021; Wang et al., 2022b), we select $k$ most relevant samples from the train set (so, those with higher BM25 scores) for each test sentence in the experiments.

**Influence** functions (Koh and Liang, 2017) were originally used in statistics for the context of linear

model analysis (Cook and Weisberg, 1982; Chatterjee and Hadi, 1986; Hampel et al., 1986). Koh and Liang (2017) adapt the functions for machine learning (ML) to understand model behaviour, debug models, detect dataset errors, and create adversarial training samples. The aim of the functions is to calculate the influence of a training sample $s^{train}$ on a test sample $s^{test}$, formulated as the change in loss on $s^{test}$, if the training sample $s^{train}$ were removed from training. This yields the influence of $s^{train}$ to solve the task for $s^{test}$.

The influence method is used in the literature to detect errors in the dataset and to create adversarial training samples (Koh and Liang, 2017). We adapted Influence as a method to study potential performance gains for ICL sample selection because it scores the contribution of a sample to the training process. Similar to in-context sample selection methods, we select $k$ samples from the train set that have a higher influence on sentences from the test set by using the baseline finetuned RoBERTa model (see Section 4.2) as the model to calculate the loss in the experiments. Since the Influence method's practical applicability is limited (it uses test labels to select the ICL samples via the loss), we use it as a best-case (or oracle) baseline, where the sample ranking is based on training utility, rather than a vocabulary similarity signal.

## 4 Experiments

### 4.1 Datasets

We evaluate the sample selection methods in ICL for entity extraction from scientific documents. We use 5 datasets from the different subdomains:

- **ADE** (Gurulingappa et al., 2012): a subset of MEDLINE case reports describing adverse effects arising from drug use.

- **MeasEval**[2] (Harper et al., 2021): a dataset collected from scientific documents from 10 different subjects and annotated for 4 entity types (Quantity, Measured Property, Measured Entity, Qualifier). Since the other entities are dependent (e.g., triggered or nested) on quantity entities, we use only `Quantity` entity type in our experiments.

- **SciERC**[3] (Luan et al., 2018): an extension of SemEval 2017 Task 10 (SemEval 17) (Augenstein et al., 2017) and SemEval 2018 Task 7 (SemEval 18) (Buscaldi et al., 2017) datasets. The dataset contains 500 abstracts of Artificial Intelligence (AI) papers with 6 scientific entity types[4].

- **STEM-ECR**[5] (D'Souza et al., 2020): a dataset containing abstracts from the same subjects of MeasEval dataset for scientific entity extraction, classification, and resolution. Although there are 7 entity types, we follow the baseline study (D'Souza et al., 2020) and use 4 of them: `Data`, `Material`, `Method`, and `Process`.[6]

- **WLPC** (Kulkarni et al., 2018): a dataset collected from wet lab protocols for biology and chemistry experiments providing entity, relation, and event annotations for wet lab protocols.

Statistical details of datasets are given in Table 1.

## 4.2 Baseline Methods

In our experiments, we compare ICL sample selection methods with a finetuned pre-trained language model, RoBERTa, zero-shot learning in which no samples are used for the GPT-3.5 prompt, and random sampling in which samples are randomly selected for the prompt.

---

[2] https://github.com/harperco/MeasEval

[3] http://nlp.cs.washington.edu/sciIE/

[4] We use `Other` as the shortened form of `OtherScientificTerm` in the rest of the paper

[5] https://data.uni-hannover.de/dataset/stem-ecr-v1-0

[6] We thus leave out Task, Object, and Results entity types, since these are almost always nested within the other scientific entity types.

**Finetuned RoBERTa baseline** To compare the sample selection methods in ICL against a sensible baseline, we trained an entity extraction model on the datasets using RoBERTa (Liu et al., 2019) PLM (RoBERTa-base). We formulate the fully tuned task as token-level labelling using the BIO tags.

**Zero-Shot** For zero-shot setup, we formulate prompts using only format (I; see Appendix B) and test sentences from the test sets for each dataset.

**Random Sampling** In this approach, we randomly select $k$ in-context samples from the train set for every test sentence.

## 4.3 Experimental Setup

Baseline RoBERTA PLM is finetuned utilising Hugging Face[7] (Wolf et al., 2020) library. The hyperparameters used in the finetuning PLM are the batch size of 32, max length of 128, the learning rate of 1e-5, and 15 epoch of training, and experiments are done on a single NVIDIA Quadro RTX 5000 GPU. We train the model five times with different random seeds and report the mean and standard deviation of the results to account for the training variance of the model.

For the baseline, zero-shot and random sampling, and ICL sample selection experiments, we build the system using the EasyInstruct[8] (Ou et al., 2023) framework to instruct LLMs for entity extraction from scientific documents with defined entity extraction prompts and entities of the datasets. In the experiments for ICL sample selection, we use a maximum of 20 in-context samples due to the GPT-3 (*gpt-3.5-turbo-0301*) token limit and 100 sentences from each test set because of the cost of GPT-3.5 usage. The experiment is repeated five times on the test set to calculate the average score and corresponding standard deviation for random sampling (see detailed results in Appendix D).

For the KATE, we use `[CLS]` token embeddings of the RoBERTa PLM and OpenAI embedding API (*text-embedding-ada-002*) to obtain sentence embeddings. We treat the embedding generation method (RoBERTa vs. GPT) as another hyperparameter (much like the number of samples $k$). We calculate the distance between embeddings using the Euclidean and cosine similarity metrics for each test sentence and select similar $k$ sentences based on the distance scores in KATE. We calculate the

---

[7] https://huggingface.co/

[8] https://github.com/zjunlp/EasyInstruct

| Method | ADE | MeasEval | SciERC | STEM-ECR | WLPC |
|---|---|---|---|---|---|
| *Baseline models* | | | | | |
| RoBERTa | $\mathbf{90.42}_{\pm0.13}$ | $\mathbf{50.68}_{\pm3.93}$ | $\mathbf{68.52}_{\pm1.30}$ | $\mathbf{69.70}_{\pm3.46}$ | $28.36_{\pm11.25}$ |
| Zero-shot | 71.29 | 19.65 | 17.86 | 28.89 | 31.64 |
| Random | $74.56_{\pm0.33}$ | $22.49_{\pm1.45}$ | $29.27_{\pm0.73}$ | $26.85_{\pm1.26}$ | $32.20_{\pm1.22}$ |
| *In-context sample selecting methods* | | | | | |
| KATE | 83.11 ‡ | 22.75 | 29.97 | 30.78 ‡ | 45.02† ‡ |
| Perplexity | 79.13 ‡ | 21.43 | 31.31 | 26.57 | 30.46 † |
| BM25 | 77.28 ‡ | 24.72 ‡ | 35.96 ‡ | 25.61 | 44.14 † ‡ |
| Influence | 86.35 ‡ | 27.13 ‡ | 36.47 ‡ | 27.81 † ‡ | **45.41** † ‡ |

Table 2: Main results for methods of selecting in-context samples. The best results are given in **bold**. The best results of the in-context sample selection method are given in underline.
† denotes statistical significance level of $p = 0.05$ compared to the supervised RoBERTa baseline and ‡ denotes statistical significance level of $p = 0.05$ compared to the random sampling.
The entity-level Macro $F_1$ score of datasets on the full test set are for ADE $89.00_{\pm0.07}$, MeasEval $65.62_{\pm5.54}$, SciERC $62.59_{\pm0.11}$, STEM-ECR $66.43_{\pm0.42}$, and WLPC $40.51_{\pm0.32}$ .

| Method | ADE | MeasEval | SciERC | Stem-ECR | WLPC |
|---|---|---|---|---|---|
| RoBERTa$_{full}$ | $90.42_{\pm0.13}$ | $50.68_{\pm3.93}$ | $68.52_{\pm1.30}$ | $69.70_{\pm3.46}$ | $28.36_{\pm11.25}$ |
| *Baseline models* | | | | | |
| RoBERTa$_{\%1}$ | $14.32_{\pm71.09}$ | $19.20_{\pm12.90}$ | $10.16_{\pm0.30}$ | $15.42_{\pm5.78}$ | $10.37_{\pm0.50}$ |
| Zero-shot | 71.29 | 19.65 | 17.86 | **28.89** | 31.64 |
| Random$_{\%1}$ | $66.53_{\pm0.19}$ | $21.32_{\pm0.88}$ | $25.31_{\pm0.66}$ | $21.38_{\pm1.89}$ | $28.46_{\pm1.77}$ |
| *In-context sample selecting methods* | | | | | |
| KATE$_{\%1}$ | 69.06† ‡ | **24.48** † ‡ | 26.78† | 26.49 † ‡ | 28.97 † |
| Perplexity$_{\%1}$ | 68.83† ‡ | 22.23† | 26.42† | 25.84† ‡ | 26.05† |
| BM25$_{\%1}$ | 72.66† ‡ | 23.39† ‡ | 31.33† ‡ | 24.24† ‡ | **36.73**† ‡ |
| Influence$_{\%1}$ | **73.68**† ‡ | 24.21 † ‡ | 32.49 † ‡ | 25.01 † ‡ | 34.24 † ‡ |

Table 3: Main results for methods of selecting in-context samples using %1 of train set. The best results are given in **bold**. The best results of the in-context sample selection method are given in underline.
† denotes statistical significance level of $p = 0.05$ compared to the supervised RoBERTa baseline (RoBERTa$_{\%1}$) and ‡ denotes statistical significance level of $p = 0.05$ compared to the random sampling (Random$_{\%1}$) for low-resource scenario.

perplexity of the samples from the train set by using the RoBERTa PLM (using the method outlined in (Salazar et al., 2019)) and select $k$ samples with the lowest perplexity for all test sets of the datasets in the Perplexity method. For BM25, we utilise rank-bm25[9] library with default parameters (term frequency saturation - k1 of 1.5, document length normalisation - b of 0.75, and constant for negative IDF of a sentence in the data - $\epsilon$ of 0.25). We use the finetuned RoBERTa to select $k$ samples, as defined in the study of Jain et al. (2022), for each test sentence in the Influence method.

As the evaluation metric, we use entity-level Macro $F_1$ score.

**Statistical significance** The statistical significance of differences in macro $F_1$ score is evaluated with an approximate randomisation test (Chinchor, 1992) with $99,999$ iterations and significance level $\alpha = 0.05$ for sample selection methods (KATE, Perplexity, BM25, and Influence) and supervised RoBERTa baseline model and the random sampling (e.g., influence → RoBERTa and influence → random sampling). For significance testing, we used the results yielding the median entity-level Macro $F_1$ score for the supervised RoBERTa baseline model and the random sampling (so, a run close to the mean value reported in the tables).

---

[9] https://pypi.org/project/rank-bm25/

## 5 Results and Discussion

### 5.1 Main Findings for Selecting In-context Samples

Our main experimental results are given in Table 2 for randomly selected 100 sentences from each of the test sets of the datasets (see Section 4.1) for entity extraction. Detailed experiments with various $k$ samples in ICL can be found in Appendix D.

Before drilling down into the in-context sample selection methods, we note that the baseline model, RoBERTa, outperforms the ICL for entity extraction from scientific documents across all datasets except WLPC, similar to the study of Gutiérrez et al. (2022) conducted on Biomedical IE. We get the highest entity-level Macro $F_1$ score among sample-selection methods for all datasets using the Influence method. Additionally, the performance of sample selection methods is low for the Measeval, SciERC, and STEM-ECR datasets, and the gap between the results of finetuned RoBERTa baseline and the Influence method is very large for these datasets. This difference in performance may be due to the difficulty of the datasets (SciERC, STEM-ECR) and the differences between train and test sets of the datasets (Measeval) (see Appendix A for a detailed analysis).

The Influence method performs comparably with the RoBERTa model for the ADE dataset. Moreover, despite the complexity of the WLPC dataset with 18 entity types, it is surprising that the effectiveness of zero-shot and ICL is better than that of the finetuned RoBERTa model. We hypothesise that this might be due to the method selecting samples from the correct minority classes. Interestingly, the textual similarity signal is almost as good, as the results of both BM25 and KATE are almost as good.

### 5.2 Low-Resource Scenario

To understand how important the size of the training set is for fully supervised finetuning of the baseline PLM model, RoBERTa, and sample selection methods for ICL, we run the experiments with 1% of the train set to simulate a low-resource scenario. The results can be found in Table 3. Although there is a decrease in the results of ICL for all datasets, it is much less drastic than for the supervised models, which is not surprising. It is well known that a sufficient amount of annotated data is needed to finetune PLM. Therefore, the robustness of ICL methods is a valuable finding that can be applied

to low-resource problems without annotated data (zero-shot) or with very small train sets (few-shot using selected samples).

### 5.3 Test Set

To understand the impact of the test set in the experiments, we used 3 different randomly sampled test sets. We present the results for the ADE and WLPC datasets (see Appendix C for statistical details of test sets), where ICL methods perform competitively with the fully supervised baseline. The results can be found in Table 4 and 5 for ADE and WLPC, respectively. It can be seen that the first test set of the WLPC dataset is challenging for the baseline model, finetuned RoBERTa. However, in-context sample selection methods, with the exception of Perplexity, appear to be less affected by the test set composition and yield similar results across different test sets.

| Method | Set 1 | Set 2 | Set 3 |
|---|---|---|---|
| *Baseline models* | | | |
| RoBERTa | **90.42**$_{\pm0.13}$ | **92.15**$_{\pm0.01}$ | **88.68**$_{\pm0.25}$ |
| Zero-shot | 71.29 | 72.87 | 72.24 |
| Random | 74.56$_{\pm0.33}$ | 72.23$_{\pm1.13}$ | 75.83$_{\pm3.15}$ |
| *In-context sample selecting methods* | | | |
| KATE | 83.11 | 84.47 | 82.65 |
| Perplexity | 79.13 | 77.31 | 77.72 |
| BM25 | 77.28 | 78.89 | 77.76 |
| Influence | 86.35 | 85.43 | 84.21 |

Table 4: Results for different test sets for ADE dataset. The best results are given in **bold**. The best results of the in-context sample selection method are given in underline.

| Method | Set 1 | Set 2 | Set 3 |
|---|---|---|---|
| *Baseline models* | | | |
| RoBERTa | 28.36$_{\pm11.25}$ | 35.93$_{\pm4.18}$ | 26.42$_{\pm1.79}$ |
| Zero-shot | 31.64 | 37.32 | 37.30 |
| Random | 32.20$_{\pm1.22}$ | 35.17$_{\pm2.25}$ | 30.77$_{\pm3.17}$ |
| *In-context sample selecting methods* | | | |
| KATE | 45.02 | **46.86** | 42.47 |
| Perplexity | 30.46 | 34.96 | 38.38 |
| BM25 | 44.14 | 41.44 | **43.20** |
| Influence | **45.41** | 41.39 | 43.18 |

Table 5: Results for different test sets for WLPC dataset. The best results are given in **bold**. The best results of the in-context sample selection method are given in underline.

### 5.4 Error Analysis

In Table 6, we give the entity-type-wise entity-level Macro $F_1$ score for the datasets for each ICL method and baseline models. The detailed error analysis of the *Influence* method – our oracle

| Dataset | Entity | Baseline Models | | | In-context Sample Selection Methods | | | |
|---|---|---|---|---|---|---|---|---|
| | | RoBERTa | Zero-shot | Random | KATE | Perplexity | BM25 | Influence |
| ADE | Adverse-Effect | 86.29 | 60.13 | 62.61 | 79.79 | 72.61 | 69.22 | 84.16 |
| | Drug | 95.48 | 82.45 | 86.12 | 86.43 | 85.65 | 85.34 | 88.54 |
| MeasEval | Quantity | 49.55 | 19.65 | 22.15 | 22.75 | 21.43 | 24.72 | 27.13 |
| SciERC | Generic | 71.43 | 5.23 | 18.42 | 18.67 | 18.32 | 26.61 | 27.11 |
| | Material | 71.88 | 6.34 | 10.21 | 17.64 | 17.55 | 20.67 | 19.45 |
| | Method | 74.14 | 45.52 | 52.18 | 52.65 | 61.26 | 62.15 | 63.08 |
| | Metric | 76.19 | 0.00 | 15.43 | 15.66 | 16.58 | 18.12 | 17.37 |
| | Other | 66.30 | 24.23 | 55.62 | 46.82 | 50.01 | 52.43 | 55.33 |
| | Task | 60.69 | 12.17 | 23.67 | 23.47 | 30.15 | 35.84 | 36.46 |
| STEM-ECR | Data | 73.71 | 29.12 | 29.33 | 32.52 | 27.61 | 23.34 | 29.45 |
| | Material | 89.21 | 36.98 | 23.18 | 31.45 | 32.18 | 31.44 | 37.12 |
| | Method | 51.61 | 22.23 | 21.12 | 29.45 | 19.16 | 22.34 | 22.27 |
| | Process | 76.17 | 31.56 | 24.41 | 29.68 | 28.28 | 27.89 | 31.11 |
| WLPC | Action | 35.84 | 69.24 | 60.11 | 72.67 | 61.23 | 71.13 | 81.24 |
| | Amount | 26.51 | 53.27 | 53.25 | 57.22 | 39.52 | 66.23 | 55.28 |
| | Concentration | 47.62 | 36.11 | 50.32 | 46.18 | 37.28 | 46.45 | 46.11 |
| | Device | 37.04 | 26.18 | 16.65 | 28.26 | 6.24 | 43.15 | 42.78 |
| | Generic-Measure | 33.33 | 0.00 | 30.88 | 0.00 | 0.00 | 0.00 | 0.00 |
| | Location | 18.18 | 21.53 | 29.35 | 38.43 | 20.00 | 49.45 | 45.22 |
| | Measure-Type | 0.00 | 0.00 | 0.00 | 0.00 | 0.00 | 0.00 | 0.00 |
| | Mention | 46.15 | 0.00 | 0.00 | 0.00 | 0.00 | 0.00 | 0.00 |
| | Method | 26.92 | 15.64 | 15.21 | 32.45 | 8.15 | 29.41 | 15.52 |
| | Modifier | 0.00 | 26.18 | 20.62 | 36.18 | 14.32 | 28.32 | 37.45 |
| | Numerical | 21.89 | 0.00 | 36.54 | 0.00 | 35.42 | 0.00 | 43.37 |
| | Reagent | 0.00 | 46.18 | 40.23 | 58.46 | 42.05 | 53.42 | 62.33 |
| | Seal | 0.00 | 0.00 | 0.00 | 0.00 | 0.00 | 0.00 | 0.00 |
| | Size | 46.15 | 0.00 | 66.96 | 0.00 | 29.32 | 60.24 | 0.00 |
| | Speed | 0.00 | 60.45 | 0.00 | 50.29 | 33.33 | 60.27 | 70.45 |
| | Temperature | 42.31 | 80.34 | 80.41 | 92.10 | 67.21 | 71.23 | 86.18 |
| | Time | 32.94 | 67.42 | 71.18 | 67.37 | 72.41 | 67.18 | 67.43 |
| | pH | 66.67 | 0.00 | 0.00 | 0.00 | 0.00 | 100 | 100 |

Table 6: Entity-type-wise results of each in-context sample selection method and baseline models.

method – shows that there are 2 types of errors in the predictions: (1) correct entity type – wrong entity span, where the model predicts an entity with correct entity type that is not annotated in the dataset, (2) wrong entity type – wrong entity span, where the model predicts an entity with a wrong entity type. The visualisation of the sample 15 sentences for error analysis can be found in Appendix E.

For the ADE dataset, all models perform better for the Drug entity type. The reason may be the shorter entity length (Adverse-Effect: 18.85, Drug: 10.27) and small vocabulary (Adverse-Effect: 2,786, Drug: 1,290), although the frequency of Adverse-Effect is higher than Drug in the train set and also in the selected samples in each in-context sample selection method. Unlike other datasets, we also encounter predictions with entity types that are not present in the ADE dataset (e.g., Disease, Number, Route).

For the MeasEval dataset, the most common error is the mislabeling of spans corresponding to other entity types (Measured Property, Measured Entity, and Qualifier, which are left out in

this study) as Quantity entities, e.g., Qualifier as Quantity (a more specific example: 'total counts per gram' predicted as Quantity, instead of the correct entity type – Qualifier). Another conclusion from the error analysis for the Measeval dataset is that GPT-3.5 tends to predict entity spans that are longer than the gold ones (e.g., gold: '11%' - predicted: 'axis 2 =11%').

Results from the SciERC dataset show that ICL with sample selection methods struggles in the prediction of less frequent entity types (Generic, Material, Metric, Task) compared to entity types with higher frequency. In particular, Other is the most frequent entity type in the dataset and GPT-3.5 often extracts a correct span and mislabels it as Other entity type. In addition, the average sentence length of SciERC is higher than the other datasets. However, the number of entities is less than the other datasets, and the Influence method tends to retrieve samples with more entities than the whole dataset. This results in extracting entities that are not actually entities in the dataset.

For the STEM-ECR dataset, the Influence method is able to extract the correct spans. How-

ever, it has difficulty in accurately labelling the spans because the dataset is imbalanced. The frequency of the Material and Process entity types is higher, which leads the Influence method to select samples with these entities and consequently label the extracted entities with these entity types.

Finally, the WLPC dataset is very dense in terms of entities in the sentences, despite the sentence length. Since the dataset is imbalanced (the entity types Action, Reagent, Amount, and Location occur more frequently than others), the Influence method retrieves samples covering these entities and, as a result, extracts mainly these entities. Moreover, the dataset is composed of instructional text and the Action entity is mostly a verb in the sentence, which is easy to extract and correctly label.

## 5.5 Discussion

In practical applications, one may not have enough annotated data to finetune PLM for a task. In such cases, it might be required to use ICL for the problem. Therefore, we explore the performance of the sample-selection methods which can be more effective in this case. First, we note that the random sampling method given in baseline methods is also competitive, especially in the low-resource scenario (see Section 5.2).

Among the sample selection methods, we obtain the best results for ADE and WLPC with sentences coming from the [CLS] token of finetuned RoBERTa (finetuned using the train set of datasets), for the SciERC, STEM-ECR, and MeasEval datasets, we obtain the best results with OpenAI embeddings for the KATE method. This may be due to the insufficient training set for these tasks since we use the embeddings from finetuned RoBERTa (which is also used as the baseline model in the study). On the other hand, using OpenAI embeddings in sample selection, despite being costly, avoids the pitfall of needing enough annotated training data to train a supervised model in order to be able to select samples for ICL (although, admittedly, even very under-trained PLM appear to be effective for sample selection; see further in this section).

We calculated the perplexity of sentences using pre-trained and finetuned RoBERTa language models for the Perplexity method, and we obtained better results using the finetuned RoBERTa, which highlights the benefits of domain-adaptation of a

language model for the entity extraction problem (but, again, points to the issue of needing a decent amount of training data to eventually train a few-shot model). The BM25 method, however, is very simple and effective for each of the datasets, without relying on any finetuned model (or any training, for that matter) for ICL sample selection.

Using these methods in selecting samples from a very limited training set (see Section 5.2) and testing on different test sets (see Section 5.3) shows that the methods are more robust compared to the baseline model, finetuned RoBERTa. In particular, our experiments in a simulated low-resource setting show that RoBERTa tuned with just 1% of the train set can be used effectively to improve ICL sample selection (e.g., via the KATE method), while performing very poorly on the actual prediction task. It is very valuable learning applicable to subdomains without annotated data or with very limited annotated datasets.

When we analyse the main results (see Table 2) and the results of the low-resource scenario (see Table 3), we find that KATE performs better in a data-poor set-up where the number of samples is severely limited. This shows that KATE has a remarkable ability to order a suboptimal subset of in-context samples. This suggests that KATE derives meaningful insights from limited data, making it a valuable method when data scarcity is a challenge. Also, BM25 offers an effective and efficient mechanism for sample selection that can be utilised in a true few-shot setup.

Another observation is that the Influence method, a classic technique from statistics, proves highly effective in selecting samples from a larger pool of samples. The method evaluates the impact of a training sample by assessing its effect on loss, typically the loss of test samples. While it is an oracle method, its high effectiveness highlights a performance gap between a loss-based signal and sample-similarity-based signal. We believe that bridging this gap is a challenge worth exploring in future research into ICL sample selection methods. However, it should be noted, that the effectiveness of Influence decreases in extreme few-shot setup, possibly due to a high training variance caused by a very small number of instances. This, in turn, highlights the robustness of KATE and BM25. BM25, as a keyword-matching method, does not require training (we used default hyperparameters in all experiments). KATE can fall back on a PLM's

ability to create text embeddings to overcome the training data scarcity, instead of relying on the loss signal produced with the under-trained layers of the model (i.e., the classification head).

## 6  Conclusion

In this paper, we explore the in-context sample selection methods for ICL entity extraction from scientific documents. Since entity extraction is a crucial step in IE from scientific documents, we analyse the methods in detail using several datasets from different subdomains, and with different entity types. The experimental results show that the baseline model, finetuned RoBERTa, still achieves the best results for this problem on 4 of 5 datasets. However, the in-context sample selection methods appear to be more robust to the train set data availability and achieve similar results to using a full train set when only a small annotated training set is used for the problem, yielding significantly better results than the baseline model in this low-resource setup.

Our work aims to extract entity spans using LLM with ICL. We focus on simple in-context sample selection methods based on similarity, perplexity, relevancy, and influence, and use GPT-3.5 as LLM in ICL. However, there are several alternative LLMs pre-trained on different domains, that could be more aligned with the task of scientific entity extraction. As future work, we hope to add a comparative dimension to our work by using these LLMs, since the ICL behaviour of LLMs can change depending on their scale and pretraining. We also plan to explore the performance of the in-context sample ordering methods (Lu et al., 2021), which are shown to impact the ICL effectiveness as well.

## Limitations

We investigate the impact of the ICL selection methods for entity extraction from scientific domains. Although we tested several methods on various datasets from different subdomains, due to the high cost of LLM models, we limited our experiments to a small subset of test sets and used only GPT-3.5. Moreover, the methods, KATE, Perplexity, and Influence (an oracle method), require finetuned models for better performance in selecting samples from the annotated dataset. In addition, we did not investigate which instruction is most appropriate. We also did not directly investigate the ordering of the selected samples, also shown to have impact of

effectiveness for related NLP problems (Lu et al., 2021; Rubin et al., 2021). Moreover, $k$ is a hyperparameter in few-shot learning that depends on the sample selection method and the dataset. We tested directly on the test set without using a validation set. Finally, we did not apply contextual calibration (Zhao et al., 2021) for entity extraction, which has been shown to improve the performance of contextual learning for NLP tasks, and kept this as future work.

## Ethics Statement

The datasets used in our experiments are publicly available. Both these datasets are focused on processing (publicly available) scientific literature, thus constituting a low-risk setting.

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

# A  Dataset Details

To understand the performance of the methods on the datasets, we calculated the difficulty of the datasets and the similarity between the train and test sets of datasets. As difficulty metrics, we use 2 metrics: Entity Ambugity Degree (EAD), and Text Complexity (TC) (Wang et al., 2022a). We also use Target Vocabulary Covered (TVC) as similarity metric (Dai et al., 2019). The details are given in Table 7.

EAD captures observable variation in the information complexity of datasets and our findings show that the SciERC and STEM-ECR datasets have the highest degree of ambiguity, implying that it is more difficult for models to predict correct

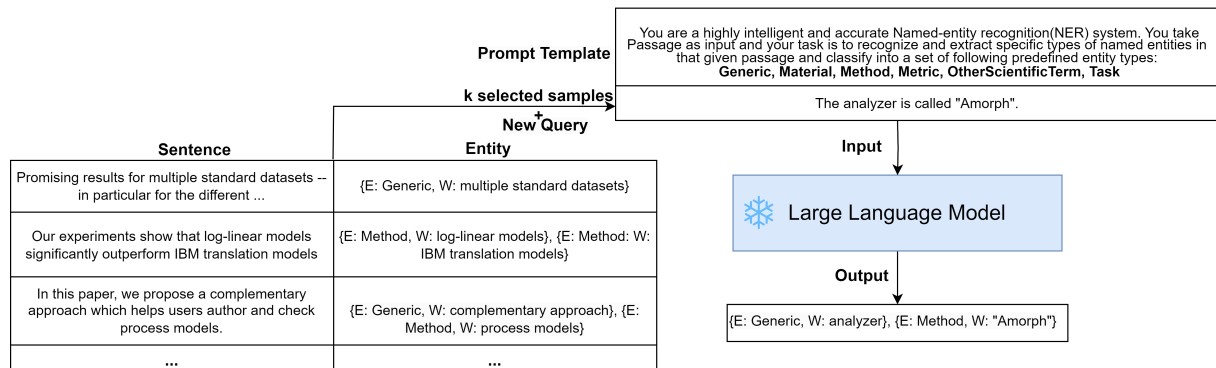

Figure 1: Illustration of in-context learning for entity extraction.

| | Difficulty | | Similarity |
|---|---|---|---|
| **Dataset** | **EAD** | **TC** | **TVC** |
| ADE | 0.42 | 30.72 | 81.51 |
| MeasEval | 0.32 | 9.13 | 53.68 |
| SciERC | 2.26 | 41.09 | 68.12 |
| STEM-ECR | 2.07 | 61.06 | 66.12 |
| WLPC | 1.51 | 35.79 | 69.32 |

Table 7: Difficulty and similarity scores of datasets.

| Dataset | Test set | # Entities | Avg $e$ | Avg $s$ |
|---|---|---|---|---|
| ADE | Set 1 | 260 | 15.12 | 133.06 |
| | Set 2 | 247 | 14.71 | 129.7 |
| | Set 3 | 227 | 15.88 | 123.4 |
| WLPC | Set 1 | 462 | 7.85 | 72.72 |
| | Set 2 | 457 | 8.39 | 68.62 |
| | Set 3 | 383 | 8.51 | 80.83 |

Table 8: Statistical details of test sets used in Section 5.3. Avg $e$ is the average length of entities and Avg $s$ is the average length of sentences.

entity types for ICL methods. It can also be seen that the TC values of the SciERC and STEM-ECR datasets are higher than those of the other datasets. In addition to the difficulty metrics, the TVC similarity metric calculates the similarity of the tokens in the training and test datasets and shows that the MeasEval test set is less similar to the train set compared to the other datasets.

## B Prompt Template

For the experiments, we use the prompt format (I) of the EasyInstruct framework defined for the Named Entity Extraction (NER) task. The prompt used in zero-shot and few-shot learning is given in Figure 1 with the illustration of ICL for entity extraction.

## C Test Set Details

Test set details used in Section 5.3 are given in Table 8.

## D In-Context Learning Experiments

The experimental results with various $k$ samples in ICL conducted for 100 sentences can be found in Table 9.

## E Visualization of Entities

The visualization of errors made by the Influence method with gold entities for 15 sentences are given in Table 10, 11, 12, 13 and 14 for ADE, MeasEval, SciERC, STEM-ECR, and WLPC datasets, respectively. We use different colours except green to highlight the entity types and we highlight the wrong entity type even if the extracted entity is correct, and the wrong extracted or wrong labeled entity with green, in the prediction of Influence method.

| Method | ADE | MeasEval | SciERC | STEM-ECR | WLPC |
|---|---|---|---|---|---|
| *Random Sampling* | | | | | |
| 1-shot | $70.62_{\pm1.23}$ | $19.99_{\pm1.88}$ | $21.10_{\pm0.01}$ | $24.27_{\pm0.23}$ | $27.18_{\pm1.22}$ |
| 3-shot | $72.24_{\pm1.09}$ | $18.51_{\pm1.69}$ | $22.34_{\pm3.25}$ | $25.87_{\pm1.25}$ | $29.34_{\pm1.22}$ |
| 5-shot | $\mathbf{74.56}_{\pm0.33}$ | $20.61_{\pm1.72}$ | $23.83_{\pm1.09}$ | $25.67_{\pm1.24}$ | $29.77_{\pm1.23}$ |
| 10-shot | $73.63_{\pm0.89}$ | $18.30_{\pm1.51}$ | $26.69_{\pm2.65}$ | $\mathbf{26.85}_{\pm1.26}$ | $\mathbf{32.20}_{\pm1.22}$ |
| 20-shot | $72.52_{\pm7.30}$ | $\mathbf{22.49}_{\pm1.45}$ | $\mathbf{29.27}_{\pm0.73}$ | $26.83_{\pm1.23}$ | $29.57_{\pm1.18}$ |
| *KATE* | | | | | |
| 1-shot | 71.44 | 20.65 | 23.94 | 26.21 | 32.34 |
| 3-shot | 77.76 | 20.29 | 24.97 | 23.89 | 43.33 |
| 5-shot | 81.45 | 21.76 | 27.56 | 26.22 | 37.06 |
| 10-shot | **83.11** | **22.75** | **29.97** | 26.60 | 40.68 |
| 20-shot | 77.31 | 22.55 | 29.84 | **30.78** | **45.02** |
| *Perplexity* | | | | | |
| 1-shot | 72.45 | 20.98 | 17.42 | 26.26 | 20.03 |
| 3-shot | 75.12 | 19.73 | 22.12 | **26.57** | **30.46** |
| 5-shot | **79.13** | 20.58 | 27.28 | 23.15 | 24.84 |
| 10-shot | 78.52 | 21.08 | **31.31** | 22.86 | 24.53 |
| 20-shot | 76.51 | **21.43** | 28.79 | 24.11 | 21.13 |
| *BM25* | | | | | |
| 1-shot | 75.40 | 21.43 | 24.42 | 24.55 | 35.01 |
| 3-shot | 75.94 | 21.37 | 28.46 | **25.61** | 38.35 |
| 5-shot | 74.99 | 23.24 | 31.90 | 24.01 | 39.66 |
| 10-shot | **77.28** | 23.76 | **35.99** | 23.69 | 42.09 |
| 20-shot | 76.74 | **24.72** | 35.96 | 24.94 | **44.14** |
| *Influence* | | | | | |
| 1-shot | 72.13 | 24.45 | 21.15 | 18.54 | 31.47 |
| 3-shot | 78.67 | 15.52 | 24.18 | 24.18 | 35.53 |
| 5-shot | **86.35** | **27.13** | 30.78 | **27.81** | 40.36 |
| 10-shot | 83.36 | 26.74 | **36.47** | 26.43 | **45.41** |
| 20-shot | 78.23 | 25.42 | 35.11 | 25.15 | 41.18 |

Table 9: ICL experiments with different $k$ in-context samples. The best results for each in-context sample selection method are given in **bold**.

| | Sentence |
|---|---|
| S1 - Gold | Gemcitabine_Drug - induced pulmonary toxicity_AE is usually a dramatic condition . |
| S1 - Influence | Gemcitabine_Drug - induced pulmonary toxicity_AE is usually a dramatic condition . |
| S2 - Gold | Peripheral neuropathy_AE associated with capecitabine_Drug . |
| S2 - Influence | Peripheral neuropathy_AE associated with capecitabine_Drug . |
| S3 - Gold | Two cases of mequitazine_Drug - induced photosensitivity reactions_AE . |
| S3 - Influence | Two cases of mequitazine_Drug - induced photosensitivity reactions_AE . |
| S4 - Gold | Captopril_Drug - induced bone marrow suppression_AE bone marrow suppression in two cardiac patients with trisomy 21 . |
| S4 - Influence | Captopril_Drug - induced bone marrow suppression_AE in two cardiac patients with trisomy 21 . |
| S5 - Gold | We conclude that ( a ) cyclophosphamide_Drug is a human teratogen_AE , ( b ) a distinct phenotype exists , and ( c ) the safety of CP in pregnancy is in ... |
| S5 - Influence | We conclude that ( a ) cyclophosphamide_Drug is a human teratogen_AE , ( b ) a distinct phenotype exists , and ( c ) the safety_AE of CP in pregnancy_AE is in ... |
| S6 - Gold | Lethal anuria_AE complicating high dose ifosfamide_Drug chemotherapy in a breast cancer patient with an impaired renal function . |
| S6 - Influence | Lethal anuria_AE complicating high dose ifosfamide_Drug chemotherapy in a breast cancer_Disease patient with an impaired renal_AE function . |
| S7 - Gold | ... developed a constellation of dermatitis_AE , fever_AE , lymphadenopathy_AE and hepatitis_AE , beginning on the 17th day of a course of oral sulphasalazine_Drug ... |
| S7 - Influence | ... developed a constellation of dermatitis_AE , fever_AE , lymphadenopathy_AE and hepatitis_AE , beginning on the 17th day of a course of oral sulphasalazine_Drug ... |
| S8 - Gold | ... of agranulocytosis_AE and neutropenic sepsis_AE secondary to carbimazole_Drug with recombinant human granulocyte colony stimulating factor ( G - CSF ) . |
| S8 - Influence | ... of agranulocytosis_AE and neutropenic sepsis_AE secondary to carbimazole_Drug with recombinant human granulocyte colony stimulating factor_Drug ... |
| S9 - Gold | According to the literature , chlorambucil_Drug central nervous toxicity_AE is found almost exclusively in childhood nephrotic syndrome . |
| S9 - Influence | According to the literature , chlorambucil_Drug central nervous toxicity_AE is found almost exclusively in childhood nephrotic syndrome_Disease . |
| S10 - Gold | Two patients with rheumatoid arthritis developed evidence of hepatotoxicity_AE while receiving D - penicillamine_Drug . |
| S10 - Influence | Two_Number patients with rheumatoid arthritis_Disease developed evidence of hepatotoxicity_AE while receiving D - penicillamine_Drug . |
| S11 - Gold | Fulminant metoclopramide_Drug induced neuroleptic malignant syndrome_AE rapidly responsive to intravenous dantrolene . |
| S11 - Influence | Fulminant metoclopramide_Drug induced neuroleptic malignant syndrome_AE rapidly responsive to intravenous_Route dantrolene_Drug . |
| S12 - Gold | ... were performed in a patient with definite seronegative rheumatoid arthritis who developed hypogammaglobulinemia_AE in the course of ... |
| S12 - Influence | ... were performed in a patient with definite seronegative rheumatoid arthritis_Disease who developed hypogammaglobulinemia_AE in the course of ... |
| S13 - Gold | Massive subfascial hematoma_AE after alteplase_Drug therapy for acute myocardial infarction . |
| S13 - Influence | Massive subfascial hematoma_AE after alteplase_Drug therapy for acute myocardial infarction_Disease . |
| S14 - Gold | Bronchiolitis obliterans with organizing pneumonia_AE after rituximab_Drug therapy for non - Hodgkin 's lymphoma . |
| S14 - Influence | Bronchiolitis obliterans with organizing pneumonia_AE after rituximab_Drug therapy for non - Hodgkin 's lymphoma_Disease . |
| S15 - Gold | Transient trazodone_Drug - induced hypomanic symptoms_AE occurred in three depressed patients . |
| S15 - Influence | Transient trazodone - induced hypomanic symptoms_AE occurred in three depressed_Disease patients . |

Table 10: Selected sentences from the test set with gold and predicted entities for the ADE dataset. AE is the abbreviation of Adverse-Effect entity type.

| | |
|---|---|
| S1 - Gold | This scenario may also explain the other peaks in Apectodinium at $2619.6$ and $2614.7\text{ m}_{Quantity}$ (although see Section 4.1). |
| S1 - Influence | This scenario may also explain the other peaks in Apectodinium at $2619.6_{Quantity}$ and $2614.7\text{ m}_{Quantity}$ |
| S2 - Gold | In all $30_{Quantity}$ programs, the Low setting yields larger slices compared to the High setting. |
| S2 - Influence | In all $30\text{ programs}_{Quantity}$, the Low setting yields larger slices compared to the High setting. |
| S3 - Gold | Fig. 5 shows the average slice size deviation when using the lower $\text{two}_{Quantity}$ settings compared to the highest. |
| S3 - Influence | Fig. 5 shows the average slice size deviation when using the lower two settings compared to the highest. |
| S4 - Gold | We also found evidence of super-large clusters: $40\%_{Quantity}$ of the programs had a dependence cluster that consumed $\text{over half}\%_{Quantity}$ of the program. |
| S4 - Influence | We also found evidence of super-large clusters: $40\%_{Quantity}$ of the programs had a dependence cluster that consumed $\text{over half}\%_{Quantity}$ of the program. |
| S5 - Gold | The average size of the programs studied was $20\text{KLoC}\%_{Quantity}$, so these clusters of $\text{more than }10\%_{Quantity}$ denoted significant portions of code. |
| S5 - Influence | The average size of the programs studied was $20\text{KLoC}\%_{Quantity}$, so these clusters of more than $10\%_{Quantity}$ denoted significant portions of code. |
| S6 - Gold | ... but the competition between these effects results in the fracture energy being independent of the test temperature $\text{between -55 °C and -109 °C}_{Quantity}$. |
| S6 - Influence | ... but the competition between these effects results in the fracture energy being independent of the test temperature between -55 °C and -109 °C. |
| S7 - Gold | We will illustrate our tests of Liouville's theorem using data for electrons with an energy $E \approx 90\text{ keV}_{Quantity}$ and a pitch angle $\alpha \approx 170°_{Quantity}$ before they encounter Rhea. |
| S7 - Influence | We will illustrate our tests of Liouville's theorem using data for electrons with an energy $E \approx 90\text{ keV}_{Quantity}$ and a pitch angle $\alpha \approx 170°_{Quantity}$ before they encounter Rhea. |
| S8 - Gold | ... ( $10\text{-}5\text{ mbar}_{Quantity}$ ) and latitude $78°_{Quantity}$ from simulations R1–R18 (Table 1) are shown in the upper panel of Fig. 12 as a function of $10\text{ keV}_{Quantity}$ electron ... |
| S8 - Influence | ... ( $10\text{-}5\text{ mbar}_{Quantity}$ ) and latitude $78°_{Quantity}$ from simulations R1–R18 (Table 1) are shown in the upper panel of Fig. 12 as a function of $10\text{ keV}_{Quantity}$ electron ... |
| S9 - Gold | While the values are based on equinox simulations, we found seasonal differences to be insignificant, generating temperature changes of $\leq 10\text{ K}_{Quantity}$. |
| S9 - Influence | While the values are based on equinox simulations, we found seasonal differences to be insignificant, generating temperature changes of $\leq 10\text{ K}$. |
| S10 - Gold | ... velocity of the ISM with respect to Earth is $-6.6\text{ km s-1}_{Quantity}$ and the effective thermal velocity along the LOS to the star is $12.3\text{ km s-1}_{Quantity}$ (Wood et al., 2005). |
| S10 - Influence | ... velocity of the ISM with respect to Earth is $-6.6_{Quantity}$ km s-1 and the effective thermal velocity along the LOS to the star is $12.3_{Quantity}$ km s-1 (Wood et al., 2005). |
| S11 - Gold | The model profiles were convolved to a spectral resolution of R = $17,500_{Quantity}$. |
| S11 - Influence | The model profiles were convolved to a spectral resolution of R = 17,500. |
| S12 - Gold | ... over a large corpus of C code was that $89\%_{Quantity}$ of the programs studied contained at least $\text{one}\%_{Quantity}$ dependence cluster composed of $10\%_{Quantity}$ ... |
| S12 - Influence | ... over a large corpus of C code was that $89\%_{Quantity}$ of the programs studied contained at least one dependence cluster composed of $10\%_{Quantity}$ ... |
| S13 - Gold | ... of $0.18\text{ g CO2 m-2 h-1contributed}_{Quantity}$ the larger fraction of RS, $56\%_{Quantity}$, while the heterotrophic component flux of $0.15\text{ g CO2 m-2 h-1accounted}_{Quantity}$ ... |
| S13 - Influence | ... of $0.18_{Quantity}$ g CO2 m-2 h-1contributed the larger fraction of RS, $56_{Quantity}$ %, while the heterotrophic component flux of $0.15_{Quantity}$ g CO2 m-2 h- 1accounted ... |
| S14 - Gold | After fracture at $20\text{ °C}_{Quantity}$, the plastic zone at the tip of the sub-critically loaded crack was sectioned and observed using transmission optical microscopy. |
| S14 - Influence | After fracture at 20 °C, the plastic zone at the tip of the sub-critically loaded crack was sectioned and observed using transmission optical microscopy. |
| S15 - Gold | Here the rubber or thermoplastic particles are typically $\text{about }0.1\text{–}5\ \mu m_{Quantity}$ in diameter with a volume fraction of $\text{about }5\text{–}20\%_{Quantity}$. |
| S15 - Influence | Here the rubber or thermoplastic particles are typically $0.1\text{–}5\ \mu m_{Quantity}$ in diameter with a volume fraction of about $5\text{–}20\%_{Quantity}$ %. |

Table 11: Selected sentences from the test set with gold and predicted entities for the MeasEval dataset.

| | |
|---|---|
| S1 - Gold | The analyzer_Generic is called " Amorph "_Method |
| S1 - Influence | The analyzer_O is called " Amorph "_O |
| S2 - Gold | Amorph_Method recognizes NE items_O in two stages : dictionary lookup_Method and rule application_Method . |
| S2 - Influence | Amorph_Method recognizes NE items_Generic in two stages_Generic : dictionary lookup_Generic and rule application_Generic . |
| S3 - Gold | First , it_Generic uses several kinds of dictionaries_O to segment and tag Japanese character strings_O . |
| S3 - Influence | First , it uses_Method several kinds of dictionaries_O to segment_Task and tag_Task Japanese character strings_O |
| S4 - Gold | When a segment is found to be an NE item_O , this information is added to the segment and it is used to generate the final output . |
| S4 - Influence | When a segment_O is found to be an NE item_O , this information is added to the segment and it is used to generate_Method the final output_Generic . |
| S5 - Gold | Requestors can also instruct the system_Generic to notify them when the status of a request changes or when a request is complete . |
| S5 - Influence | Requestors can also instruct the system_Generic to notify_Task them when the status_O of a request_O changes or when a request_O is complete_Task . |
| S6 - Gold | This work proposes a new research direction to address the lack of structures_O in traditional n-gram models_Method . |
| S6 - Influence | This work_Generic proposes a new research direction_Method to address the lack of structures_Task in traditional n-gram models_Method . |
| S7 - Gold | Our approach_Generic is based on the iterative deformation_Task of a 3 − D surface mesh_Method to minimize an objective function_O . |
| S6 - Influence | Our approach_Generic is based on the iterative deformation_Method of a 3 − D_Metric surface mesh_O to minimize an objective function_O . |
| S8 - Gold | They_Generic improve the reconstruction_Task results and enforce their consistency with a priori knowledge_O about object shape_O . |
| S6 - Influence | They_Generic improve the_Task reconstruction_Task results and enforce their consistency with a_O priori knowledge_O about object shape_O . |
| S9 - Gold | It is based on a weakly supervised dependency parser_Task that can model speech syntax_O without relying on any annotated training corpus_Material . |
| S9 - Influence | It_Generic is based on a weakly supervised dependency parser_Method that can model speech syntax_O without relying on any annotated training corpus_Metric . |
| S10 - Gold | Labeled data_O is replaced by a few hand-crafted rules_O that encode basic syntactic knowledge_O . |
| S10 - Gold | Labeled data is replaced by a few hand-crafted rules_Method that encode basic syntactic knowledge_O . |
| S11 - Gold | The request is passed to a mobile , intelligent agent_Method for execution_Task at the appropriate database . |
| S11 - Influence | The request_Task is passed to a mobile_Generic , for execution_Task at the appropriate database_Generic . |
| S12 - Gold | Each part is a collection of salient image features_O . |
| S12 - Influence | Each part is a collection of salient image features . |
| S13 - Gold | We have conducted numerous simulations to verify the practical feasibility of our algorithm_Generic . |
| S13 - Influence | We have conducted numerous simulations_Generic to verify the practical feasibility of our algorithm_Generic . |
| S14 - Gold | In this paper , we explore what can be said about transparent objects_O by a moving observer . |
| S14 - Influence | In this paper , we explore what can be said about transparent objects_Task by a moving observer_O . |
| S15 - Gold | The result theoretically justifies the effectiveness of features_O in robust PCA_Method . |
| S15 - Influence | The result theoretically justifies the effectiveness of features_O in robust PCA_O . |

Table 12: Selected sentences from the test set with gold and predicted entities for the SciERC dataset. O is the abbreviation of Other.

Table 13: Selected sentences from the test set with gold and predicted entities for STEM-ECR dataset.

| | Sentence |
|---|---|
| S1 - Gold | FAP-specific iPS cells$_{Material}$ have potential to differentiate$_{Process}$ into hepatocyte-like cells$_{Material}$ . |
| S1 - Influence | FAP-specific iPS cells$_{Material}$ have potential to differentiate$_{Process}$ into hepatocyte-like cells$_{Material}$ . |
| S2 - Gold | Distributed source localization$_{Process}$ provided whole-brain measures$_{Data}$ from 30 to 130ms$_{Data}$ . |
| S2 - Influence | Distributed source localization$_{Method}$ provided whole-brain measures$_{Data}$ from 30 to 130ms$_{Data}$ . |
| S3 - Gold | Annealing$_{Process}$ enhances$_{Process}$ efficiency$_{Data}$ over a wide range$_{Data}$ of D:A blend compositions$_{Material}$ (1:4-4:1)$_{Data}$ . |
| S3 - Influence | Annealing$_{Process}$ enhances efficiency$_{Process}$ over a wide range$_{Data}$ of D:A blend compositions$_{Material}$ (1:4-4:1)$_{Data}$ . |
| S4 - Gold | The presented contravariant formulation$_{Data}$ is free of Christoffel symbols$_{Data}$ . |
| S4 - Influence | The presented contravariant formulation$_{Process}$ is free of Christoffel symbols$_{Material}$ . |
| S5 - Gold | Hence we recommend close monitoring$_{Process}$ of the resultant transgenic genotypes$_{Material}$ in multi-year, multi-location field trials$_{Process}$ . |
| S5 - Influence | Hence we recommend close monitoring$_{Process}$ of the resultant transgenic genotypes$_{Material}$ in multi-year, multi-location field trials$_{Data}$ . |
| S6 - Gold | Herein, we report that cobalt-substituted NaFeO2$_{Material}$ demonstrates excellent electrode performance$_{Data}$ in a non-aqueous Na cell at room temperature$_{Data}$ . |
| S6 - Influence | Herein, we report that cobalt-substituted NaFeO2$_{Material}$ demonstrates excellent electrode performance$_{Process}$ in a non-aqueous Na cell at room temperature$_{Data}$ . |
| S7 - Gold | The dual-layer carbon film$_{Material}$ is prepared using CDC process$_{Process}$ with subsequent CVD method$_{Method}$ . |
| S7 - Influence | The dual-layer carbon film$_{Material}$ is prepared$_{Process}$ using CDC process$_{Method}$ with subsequent CVD method$_{Method}$ . |
| S8 - Gold | We optimized$_{Process}$ a single-cell cryopreservation$_{Process}$ for hiPSCs in suspension$_{Material}$ . |
| S8 - Influence | We optimized a single-cell cryopreservation$_{Method}$ for hiPSCs$_{Material}$ in suspension$_{Process}$ . |
| S9 - Gold | However, no significant effects$_{Process}$ of particle size$_{Data}$ were found on the measured value of toughness$_{Data}$ . |
| S9 - Influence | However, no significant effects$_{Process}$ of particle size$_{Data}$ were found on the measured$_{Process}$ value of toughness$_{Data}$ . |
| S10 - Gold | The metal complexes$_{Material}$ exhibits different geometrical arrangements$_{Data}$ such as octahedral and square pyramidal coordination$_{Process}$ . |
| S10 - Influence | The metal complexes$_{Material}$ exhibits$_{Process}$ different$_{Data}$ geometrical arrangements$_{Data}$ such as octahedral and square pyramidal coordination$_{Data}$ . |
| S11 - Gold | ... this ion flow$_{Process}$ contributes to maintaining$_{Process}$ the nightside ionosphere$_{Material}$ near the terminator region$_{Material}$ at solar minimum$_{Data}$ . |
| S11 - Influence | ... this ion flow$_{Material}$ contributes to maintaining the nightside ionosphere$_{Process}$ near the terminator region$_{Material}$ at solar minimum$_{Data}$ . |
| S12 - Gold | ... extensive experiments$_{Method}$ are carried out on several data sets$_{Material}$ to verify$_{Process}$ the performance$_{Data}$ of the proposed algorithms$_{Method}$ . |
| S12 - Influence | ... extensive experiments$_{Process}$ are carried out on several data sets$_{Data}$ to verify the performance$_{Process}$ of the proposed algorithms$_{Process}$ . |
| S13 - Gold | Furthermore, near-homogenous populations of hFSCs$_{Material}$ can be obtained from hPSC lines$_{Material}$ which are normally ... |
| S13 - Influence | Furthermore, near-homogenous populations$_{Process}$ of hFSCs$_{Material}$ can be obtained$_{Process}$ from hPSC lines$_{Material}$ which are normally ... |
| S14 - Gold | Nodes' role-shift$_{Process}$ prevailed when a healthy network$_{Material}$ changed to diseased one$_{Material}$ . |
| S14 - Influence | Nodes' role-shift prevailed$_{Process}$ when a healthy network$_{Material}$ changed$_{Process}$ to diseased one$_{Material}$ . |
| S15 - Gold | Differences$_{Data}$ in the level$_{Data}$ of wave activity$_{Process}$ across Saturn's magnetopause$_{Material}$ has been predicted$_{Process}$ . |
| S15 - Influence | Differences in the level of wave activity$_{Data}$ across Saturn's magnetopause$_{Material}$ has been predicted$_{Process}$ . |

**S1 - Gold:** Pour out[Action] and collect[Action] the liquid[Reagent] .

**S1 - Influence:** Pour out[Action] and collect[Action] the liquid[Generic−Measure] .

**S2 - Gold:** These are the cells of interest; DO NOT DISCARD[Action] .

**S2 - Influence:** These are the cells of interest[Mention] ; DO NOT DISCARD

**S3 - Gold:** OmniPrep™ For High Quality Genomic DNA Extraction From Gram-Positive Bacteria

**S3 - Influence:** OmniPrep[Device]™ For High Quality Genomic DNA Extraction From Gram-Positive Bacteria

**S4 - Gold:** Discard[Action] the supernatant[Action] .

**S4 - Influence:** Discard[Action] the supernatant[Action] .

**S5 - Gold:** Add[Action] 450µl sterile water[Reagent] and 50µl EDTA[Reagent] to the pellet and gently[Modifier] vortex[Action] to resuspend[Action] .

**S5 - Influence:** Add[Action] 450µl[Amount] sterile[Modifier] water[Reagent] and 50µl[Amount] EDTA to the pellet and gently[Modifier] vortex[Action] to resuspend[Action] .

**S6 - Gold:** Incubate[Action] the sample[Reagent] at 55-60°C[Temperature] for 15 minutes[Time] .

**S6 - Influence:** Incubate[Action] the sample at 55-60°C[Temperature] for 15 minutes[Time] .

**S7 - Gold:** Do not heat higher than 60°C.

**S7 - Influence:** Do not heat[Action] higher than 60°C.

**S8 - Gold:** Incubate[Action] the sample[Reagent] for 5-10 minutes[Time] at 60°C[Temperature] .

**S8 - Influence:** Incubate[Action] the sample[Reagent] for 5-10 minutes[Time] at 60°C[Temperature] .

**S9 - Gold:** Add[Action] 100µl Precipitation Solution[Reagent] and mix[Action] by inverting[Action] the tube[Device] several[Numerical] times.

**S9 - Influence:** Add[Action] 100µl[Amount] Precipitation Solution[Reagent] and mix[Action] by inverting the tube[Device] several times[Modifier] .

**S10 - Gold:** Centrifuge[Action] the sample[Reagent] at 14,000xg[Speed] for 5 minutes[Time] .

**S10 - Influence:** Centrifuge[Action] the sample[Reagent] at 14,000xg[Speed] for 5 minutes[Time] .

**S11 - Gold:** Invert[Action] the tube[Device] periodically[Modifier] each hour[Time] .

**S11 - Influence:** Invert[Action] tube periodically each hour[Time] .

**S12 - Gold:** Invert[Action] the tubes[Device] 10[Numerical] times to precipitate[Action] the DNA[Reagent] .

**S12 - Influence:** Invert[Action] the lime 10[Numerical] times to precipitate[Method] the DNA[Reagent] .

**S13 - Gold:** For increased DNA recovery, add 2µl Mussel Glycogen as a DNA carrier.

**S13 - Influence:** For increased[Modifier] DNA recovery[Mention] , add[Action] 2µl[Amount] Mussel Glycogen[Reagent] as a[Amount] DNA carrier[Mention] .

**S14 - Gold:** for 1 min colorboxmagentaSpin[Action] at 1000g[Speed] for 2 minutes[Time] .

**S14 - Influence:** for 1 min[Time] colorboxmagentaSpin[Action] at 1000g[Speed] for 2 minutes[Time] .

**S15 - Gold:** Electron microscopy for virus identification and virus assemblage characterization

**S15 - Influence:** Electron microscopy[Method] for virus identification[Mention] and virus assemblage characterization[Mention]

Table 14: Selected sentences from the test set with gold and predicted entities for the WLPC dataset.