# OpenReview forum: "impact of sample selection on in-context learning for entity extraction from scientific writing"
_EMNLP/2023/Conference — EMNLP 2023 Findings_

### Official Review · Reviewer_89Vn · 2023-08-04

**Soundness:** 3

**Excitement:**

4: Strong: This paper deepens the understanding of some phenomenon or lowers the barriers to an existing research direction.

**Paper Topic And Main Contributions:**

The paper presents an analysis of In-Context Learning (ICL) sample selection methods for entity extraction from scientific documents.
A few sampling methods are used to select in-context samples from training set, namely KATE, Perplexity, BM25 and Influence. Baseline models include a finetuned RoBERTa model, as well as two LLM-based methods using zero-shot and random sampling, respectively.

The finetuned RoBERTa still achieves the best results for this problem on 4 of 5 datasets (except WLPC). However, the in-context sample selection methods appear to be more robust when there are not sufficient training samples. The paper also demonstrates that the effectiveness of in-context sample selection methods is domain-dependent, but improvements (over zero-short or random sampling) are more notable for problems with a larger number of entity types.


**Questions For The Authors:**

In Section 5.3, the description in the text and the tables do not seem to match.  You said "... in-context sample selection methods, with the exception of KATE, appear to be less affected by the test set composition and yield similar results across different test sets."
However Table 5 does not seem to suggest so?

**Reasons To Accept:**

1. This paper demonstrates methods of ICL sample selection. ICL is a new and fast developing field, and we should encourage publications in this domain.
2. It tries a variety sample selection methods, on 5 datasets. For each method, there are different choices of text embeddings too. Other researchers could adopt some of the ideas, not only for entity extraction tasks, but also for other NLP tasks involving ICL.
3. Some findings and analyses are interesting and insightful.


**Reasons To Reject:**

1. Generally speaking, the results still fall far behind the strong baseline.
2. GPT output is known to be unstable. It is unclear how consistent the evaluation results can be, with multiple runs.
3. The reasons/hypotheses why different selection methods have different results are not discussed in details, although the results are exhibited in detailed ways.

**Reproducibility:**

3: Could reproduce the results with some difficulty. The settings of parameters are underspecified or subjectively determined; the training/evaluation data are not widely available.

**Reviewer Confidence:**

4: Quite sure. I tried to check the important points carefully. It's unlikely, though conceivable, that I missed something that should affect my ratings.

---

> ### Author Rebuttal · Authors · 2023-08-29
>
> - **Generally speaking, the results still fall far behind the strong baseline.**
>
> Thank you for the comment. The main question of this study is to investigate the potential of GPT-3.5 in ICL learning for the task of entity extraction of scientific documents, without claiming that ICL is a more effective method than a fully supervised model for this task. We formulate this conclusion in Section 6.
>
>  - **GPT output is known to be unstable. It is unclear how consistent the evaluation results can be, with multiple runs.**
>
>  In our experiments, we conducted random sampling experiments multiple times (Table 2 for the main results and Table 3 for the low-resource scenario). As can be seen, the standard deviation of the random sampling method is lower than the supervised baseline model (finetuned RoBERTa). We have also run the other ICL methods several times as preliminary work for the study and have obtained similar outcomes for them (see Tables 4 and 5).  We believe that these results, while not a complete analysis of the variance, indicate that the outcomes are reasonably stable (but the results are sensitive to sample selection/order). They don’t, however, answer the questions about the stability of the model over time, as results can change with updates of a closed model. It would be interesting to see a broad reproducibility study in some months/years.
>
> - **The reasons/hypotheses why different selection methods have different results are not discussed in details, although the results are exhibited in detailed ways.**
>
> We are happy to discuss some of our hypotheses in the final version of the paper. Some of these include:
>
> * KATE works better when the choice of instances is very limited (data-poor set-up/low-resource scenario) - this leads us to believe that the method is quite good at ordering a suboptimal subset of in-context samples.
>
> * In our study, the experimental results showed that Influence is more effective in selecting samples from a larger pool of candidates. The method is originally a classic technique from robust statistics that computes the impact of a training sample by assessing its effect on loss (here, loss on the test sample).  It turns out that this signal based on sample utility is effective for ICL sample selection, possibly more effective than similarity-based signal (e.g., KATE, BM25). However, in extreme few-shot setting it appears that the loss-based signal of Influence degrades more (possibly due to high training variance with such a low number of instances) when compared to other methods (where, for example, KATE possibly falls back on PLM embedding ‘skill‘). Nonetheless, Influence still yields effectiveness significantly better than that of a random selection ICL and the supervised model, even with a candidate pool as small as a handful of instances (see experiments on MeasEval or STEM-ECR at 1% training data). Additionally, when the candidate pool becomes slightly larger (i.e., in other experiments @1% of training data), Influence yields the highest (or second highest, on one occasion) effectiveness of all ICL sample selection methods.
>
> **REFERENCES**
>
> [1] Dong, Qingxiu, et al. "A survey for in-context learning." arXiv preprint arXiv:2301.00234 (2022).
>
> [2] Koh, Pang Wei, and Percy Liang. "Understanding black-box predictions via influence functions." International conference on machine learning. PMLR, 2017.
>
> **Questions For The Authors:**
> - **In Section 5.3, the description in the text and the tables do not seem to match. You said "... in-context sample selection methods, with the exception of KATE, appear to be less affected by the test set composition and yield similar results across different test sets." However Table 5 does not seem to suggest so?**
>
> That is right, the sentence should read ‘(...) with the exception of Perplexity (...)’. Thank you for pointing this out.

---

### Official Review · Reviewer_v6zx · 2023-08-04

**Soundness:** 3

**Excitement:**

2: Mediocre: This paper makes marginal contributions (vs non-contemporaneous work), so I would rather not see it in the conference.

**Paper Topic And Main Contributions:**

This paper investigates different in-context sample selection methods for in-context learning for entity extraction from scientific documents, where annotating large datasets is expensive. It provides a comprehensive evaluation of different selection methods across different sub-domains and entity types. It finds that fine-tuned RoBERTa still outperforms different in-context sample selection methods. In-context sample selection methods, however, are robust when limited annotated data is available.

**Reasons To Accept:**

Some of the strengths of the paper are:

1. The paper provides a comprehensive analysis of in-context sample selection methods for entity extraction from scientific documents using GPT-3.5.
2. It compares the performance of in-context sample selection methods (low annotation budget) with finetuned LMs (large annotation budget), and shows the finetuned LMs still outperform LLMs with in-context learning.
3. The paper provides detailed analyzes and discussion of how different in-context sample selection methods affect performance. These findings are important to improve methods for extracting important information from scientific documents.

**Reasons To Reject:**

Some of the weaknesses of the paper are:

1. The paper only analyzes existing approaches for in-context sample selection for entity extraction from scientific documents. Findings are interesting but no new methods are proposed.
2. The findings from the paper are very similar to https://arxiv.org/abs/2203.08410. Finetuned LMs achieve superior performance in specialized domains (such as biomedical) than LLMs with in-context learning. This paper extends that study to further analyze different in-context sample selection strategies on scientific documents.
3. It would strengthen the paper if more tasks over scientific documents (e.g. relation extraction, classification etc.) are considered.
4. The paper doesn't compare against other few-shot entity extraction approaches. It will be nice to compare different approaches for low-resource settings.
5. It only limits to GPT3.5 and does not study the impact of selecting the appropriate instruction and order of in-context examples. It will be interesting to see how this behavior generalizes to other LLMs and settings.
6. Yet another direction that many recent works explore is using LLMs as annotators. That could provide annotated data to finetune smaller LMs.

**Reproducibility:**

3: Could reproduce the results with some difficulty. The settings of parameters are underspecified or subjectively determined; the training/evaluation data are not widely available.

**Reviewer Confidence:**

3: Pretty sure, but there's a chance I missed something. Although I have a good feel for this area in general, I did not carefully check the paper's details, e.g., the math, experimental design, or novelty.

---

> ### Author Rebuttal · Authors · 2023-08-29
>
> Thank you for the reviews.
>
> - **The paper only analyzes existing approaches for in-context sample selection for entity extraction from scientific documents. Findings are interesting but no new methods are proposed.**
>
> The influence method is used in the literature to detect errors in the dataset and to create adversarial training samples [1]. We adapted Influence as an ICL method for sample selection because it involves the scoring of a training sample, which allows for a straightforward application. To the best of our knowledge, we are the first to use Influence as an ICL method for sample selection. We will clarify this point in the camera-ready version.
>
> - **The findings from the paper are very similar to https://arxiv.org/abs/2203.08410. Finetuned LMs achieve superior performance in specialized domains (such as biomedical) than LLMs with in-context learning. This paper extends that study to further analyze different in-context sample selection strategies on scientific documents.**
>
> We use a somewhat similar experimental setup, but our findings are quite different. Our results show that ICL yields better results in training-data-poor setup. We agree that the focus is also different (we do focus on different ICL strategies and comparing them), but we can’t see how this broader point makes for a reason to reject our submission.
>
> - **It would strengthen the paper if more tasks over scientific documents (e.g. relation extraction, classification etc.) are considered.**
>
> We agree that extending the analysis of ICL techniques to other tasks would be interesting, but we can’t see how one could fit another 2-3 tasks, and consequently another 10+ datasets, and corresponding discussions into a single conference paper. We are happy to take this suggestion onboard to guide our future work.
>
> - **The paper doesn't compare against other few-shot entity extraction approaches. It will be nice to compare different approaches for low-resource settings.**
>
>  We position our work within the fine-tuning vs ICL discussion, but, while a search for an alternative low-resource solution lies out of the scope of this work, we agree that another few-shot baseline could be beneficial to contextualise our results. If the Reviewer pointed us to an established baseline, they’d be interested in seeing here, we’d be happy to include it in the camera-ready version of the paper.
>
> - **It only limits to GPT3.5 and does not study the impact of selecting the appropriate instruction and order of in-context examples. It will be interesting to see how this behavior generalizes to other LLMs and settings.**
>
> We agree with this comment, but due to our limited resources we only used GPT-3.5 in our experiments and state this as a limitation. On the point about order of the sample: while we do not tackle this problem explicitly, the ICL methods do yield an ordering of the samples. In fact, it probably bears heavily on the results in extreme few-shot settings (i.e., at 1% of training data for the smallest datasets), where KATE outperforms Influence. We’d be happy to elaborate on this point in the camera-ready version of the paper.
>
>  - **Yet another direction that many recent works explore is using LLMs as annotators. That could provide annotated data to finetune smaller LMs.**
>
> Thank you for the comment. Our focus here is on ICL sample selection, but using LLMs as annotators is an interesting idea.
>
> **REFERENCES**
>
> [1] Koh, Pang Wei, and Percy Liang. "Understanding black-box predictions via influence functions." International conference on machine learning. PMLR, 2017.
>
> [2] Gutierrez, Bernal Jimenez, et al. "Thinking about gpt-3 in-context learning for biomedical ie? think again." arXiv preprint arXiv:2203.08410 (2022).

---

### Official Review · Reviewer_JH68 · 2023-08-12

**Soundness:** 3

**Excitement:**

3: Ambivalent: It has merits (e.g., it reports state-of-the-art results, the idea is nice), but there are key weaknesses (e.g., it describes incremental work), and it can significantly benefit from another round of revision. However, I won't object to accepting it if my co-reviewers champion it.

**Missing References:**

1.  SPECTER: Document-level Representation Learning using Citation-informed Transformers. ACL'20.

2.  Neighborhood Contrastive Learning for Scientific Document Representations with Citation Embeddings. EMNLP'22.

3. S2ORC: The Semantic Scholar Open Research Corpus. ACL’20.

4. SciREX: A Challenge Dataset for Document-Level Information Extraction

**Paper Topic And Main Contributions:**

This work  presents a comprehensive analysis of in-context sample selection methods for entity extraction from scientific documents using GPT-3.5 and compare these results against a fully supervised transformer-based baseline.

**Reasons To Accept:**

1. Paper is well written with Good motivation.

2. Novelty of the work is strong enough.

2. Empirical results demonstrate that the in-context learning technique is really helpful without developing any other supervised setup.

**Reasons To Reject:**

1.Why only one LLM has been used rather than using other instruction based models such as  FLanT5, LLAMA and the Scientific LLM i.e., Galactica?

2. Description of Prompt templates is not provided.

3. Why authors finetuned Roberta on scientific information extraction task as the supervised baseline. There are some more advanced scientific pre-trained language models (e.g., SPECTER [1] and SciNCL [2]) which are not examined. Because RoBerta is pretrained on wikipedia dataset.

4. Usage of external information for in-context learning technique is another interesting paradigm. So, the author could use S2ORC [3] as the required external source of information.

5. SciREX[4] could be another dataset to conduct these experiments

6. The results reported in the Tables are not analyzed properly. Authors could analyze the results rather than reporting it in tabular form.

7. Significance tests are missing. To show the proposed strategy is beneficial.

8. Anonymous source code repo is not provided.

**Reproducibility:**

2: Would be hard pressed to reproduce the results. The contribution depends on data that are simply not available outside the author's institution or consortium; not enough details are provided.

**Reviewer Confidence:**

3: Pretty sure, but there's a chance I missed something. Although I have a good feel for this area in general, I did not carefully check the paper's details, e.g., the math, experimental design, or novelty.

---

> ### Author Rebuttal · Authors · 2023-08-29
>
> Thank you for the reviews.
>
> - **Why only one LLM has been used rather than using other instruction based models such as FLanT5, LLAMA and the Scientific LLM i.e., Galactica?**
>
> Thank you for the comment. We agree that a comparison of different LLM would be interesting, but due to our limited resources we only used GPT-3.5 in our experiments and stated this as a limitation of the study.
>
> - **Description of Prompt templates is not provided.**
>
> Description of prompt templates is provided in Section 3.1, Appendix B, and in Figure 1 of Appendix B. We are going to provide code repository for the experiments (on acceptance), which will also allow another way to access the prompt template. We would be happy to add an example of a full prompt to Appendix B in the camera-ready version of the paper, if the reviewers deem it beneficial to the paper.
>
> - **Why authors finetuned Roberta on scientific information extraction task as the supervised baseline. There are some more advanced scientific pre-trained language models (e.g., SPECTER [1] and SciNCL [2]) which are not examined. Because RoBerta is pretrained on wikipedia dataset.**
>
> Our intention was to provide >>a<< baseline well established in entity extraction tasks. The paper is focused on instance selection for ICL, not on comparing different PLMs in fully supervised settings. This being said, we agree that choice of a strong baseline is important to provide appropriate context. In this sense, we believe that RoBERTa works reasonably well. Our follow-up experiment involving RoBERTa and SciNCL shows that RoBERTa outperforms SciNCL on 3/5 datasets with full training data and 2/5 datasets under the limited training data regime. In any case, using a science-specific PLM, would have changed nothing in our main conclusions, SciNCL is better/worse than ICL in the same cases (dataset/data availability) as RoBERTa. While we believe that adding a science-specific model (such as SciNCL) does not add much to the discussion of ICL results, we’d be open to adding this baseline to the final version of the paper to demonstrate that the comparison between full tuning and ICL is not unique to RoBERTa.
>
> - **Usage of external information for in-context learning technique is another interesting paradigm. So, the author could use S2ORC [3] as the required external source of information.**
>
> Thank you very much for the suggestion. The aim of the study is providing a comprehensive analysis of ICL methods to choose samples from the train sets as formulated in the literature [1,2,3,4]. Therefore, however interesting, the use of external information for in-context learning (or response grounding) is out of the scope of this paper.
>
> - **SciREX[4] could be another dataset to conduct these experiments**
>
> Thank you for pointing out the dataset. In our experiments, we selected 5 datasets with different number of entities and subdomains to provide a comprehensive analysis of the ICL methods by covering different scientific subdomains. We believe that our selection is broad enough to justify the findings, and that missing the SciREX dataset does not compromise the soundness of our evaluation. Nonetheless, we’d be happy to include the SciREX dataset in our future work.
>
> - **The results reported in the Tables are not analyzed properly. Authors could analyze the results rather than reporting it in tabular form.**
>
> We are not quite sure what is improper in the analysis of the results presented in the manuscript. If there is a specific valuable insight that can be gathered from the tables, but is omitted in the discussion, we’d appreciate if the Reviewer specified it.
>
> - **Significance tests are missing. To show the proposed strategy is beneficial.**
>
> Thank you for reminding us about the significance tests. We did the tests but inadvertently forgot to include the results of the significance tests in the paper. We will add the results to the camera-ready version. Most of the ‘clear’ differences (between our approach, influence, and random baseline/trained supervised RoBERTa baseline) in the Tables 2 and 3 are also significant (we ran a pseudo randomisation test with an 0.05 confidence level), so our key conclusions stand. As for the results that already are in the manuscript (although it’s not quite the same thing), results reported in Tables 4 and 5 (as well as the variance reported on the random baseline) shed some light on the variance/confidence intervals.
>
> - **Anonymous source code repo is not provided.**
>
> As we stated in the paper, we will publish our code in the camera-ready version. We believe this would alleviate the reproducibility issues pointed out by the Reviewer.
>
> **REFERENCES**
>
> [1] Radford, Alec, et al. "Language models are unsupervised multitask learners." OpenAI blog 1.8 (2019): 9.
>
> [2] Brown, Tom, et al. "Language models are few-shot learners." Advances in neural information processing systems 33 (2020): 1877-1901.
>
> [3] Liu, Jiachang, et al. "What Makes Good In-Context Examples for GPT-3?." arXiv preprint arXiv:2101.06804 (2021).
>
> [4] Min, Sewon, et al. "Rethinking the role of demonstrations: What makes in-context learning work?." arXiv preprint arXiv:2202.12837 (2022).

---

### Meta-Review · Area_Chair_egmG · 2023-09-06

**Recommendation:** 5
**Best Paper Recommendation:** No

**Metareview:**

The paper investigates methods for in-context-learning (ICL) sample selection for entity extraction from scientific documents and compares it against a fully supervised baseline. The reviewers acknowledge that the motivation for the work is important and that the experiments demonstrate the key contribution, that ICL is useful for low-resource settings even though a fully-supervised solution still outperforms ICL (which is to be expected).

The majority of the issues identified by the authors are suggestions for further research (e.g. considering more NLP tasks) or changes to the experimental setup that don't affect the conclusions of the work (e.g. using a stronger supervised baseline). The remaining issues seem to have been successfully addressed by the authors.

---

### Decision · Program_Chairs · 2023-10-07

**Decision:**

Accept-Findings

**Comment:**

The paper investigates methods for in-context-learning (ICL) sample selection for entity extraction from scientific documents and compares it against a fully supervised baseline. The reviewers acknowledge that the motivation for the work is important and that the experiments demonstrate the key contribution, that ICL is useful for low-resource settings even though a fully-supervised solution still outperforms ICL (which is to be expected).

The majority of the issues identified by the authors are suggestions for further research (e.g. considering more NLP tasks) or changes to the experimental setup that don't affect the conclusions of the work (e.g. using a stronger supervised baseline). The remaining issues seem to have been successfully addressed by the authors.